# Effect of Abrasive Grain Size on the Abrasion Volume Loss of Subfossil and Recent Oak Wood in Three Characteristic Sections

**DOI:** 10.3390/ma16010432

**Published:** 2023-01-03

**Authors:** Sara Essert, Vera Rede, Josip Barišić

**Affiliations:** 1Division of Botany, Department of Biology, Faculty of Science, University of Zagreb, 10000 Zagreb, Croatia; 2Department of Materials, Faculty of Mechanical Engineering and Naval Architecture, University of Zagreb, 10000 Zagreb, Croatia; 3Bestprojekt d.o.o., 10000 Zagreb, Croatia

**Keywords:** abonos, abrasive resistance, grit size, pedunculate oak, wood anisotropy

## Abstract

Subfossil wood is a valuable and rare material often used for production of expensive furniture and decorative artistic items of unique beauty. Its mechanical and tribological properties are still being studied and are considered specific due to the particular conditions of its long-lasting formation in aqueous sediment sludge. Various elements that have been impregnated into the wood tissue over many years make the machining and grinding of this type of wood rather difficult compared to normal recent wood. The main objective of this study was to determine the influence of the abrasive grain size of sandpaper on the abrasion volume loss of recent and two subfossil oak samples in three characteristic sections (cross, radial, and tangential). The results showed that the average size of abrasive grains and the orientation of the wood structure have an influence on the abrasion volume loss of all three samples. The phenomenon of the critical size of abrasive grains was observed in all samples and on all sections. As the size of abrasive grains increased to the critical size, the abrasive volume loss of the sample increased simultaneously. The lowest abrasion volume loss was observed on recent oak. In all samples, the lowest volume loss was measured on the cross sections, and the tangential and radial sections had mutually equal values. It was also found that the increase in the size of abrasive grains to a critical value resulted in the increasing value of the absolute difference between the abrasion volume loss of the cross, radial, and tangential section samples, while the relative relations between the abrasive volume loss values of three different sections (C/R, C/T, R/T) within the same grit of sandpaper remained quite similar.

## 1. Introduction

Besides stone, wood was the first material used by man for its being a high-quality engineering material, and owing to its unique properties it is still widely used today. Wood is a primarily natural, renewable, fully recyclable and technically versatile material. As such, it is largely used both in its natural state and as processed wood [1].

If wood remains buried for a long time in the soil under the influence of flowing or stagnant water, it becomes subfossil wood, also called abonos or bog wood. The water and the sludge in which the wood has been immersed for many years create anoxic conditions not favourable for the development of fungi and micro-organisms that would cause biodegradation. Increased durability and changes in the colour and mechanical properties of subfossil wood are caused by the deposition of minerals from flowing water and surrounding mud [2]. There are a number of studies dealing with the chemical and physical properties of subfossil wood [3]. The most frequently investigated have been the chemical properties of subfossil wood [4,5,6,7] and the influence of abiotic and biotic factors in the aquatic environment on decomposition and changes in the wood [8,9,10]. Since subfossil wood is a rare but valuable building material, numerous physical and mechanical properties have also been investigated, such as bending modulus [11], pressure resistance [12], bending strength [13], density [14,15], compression strength [14], static hardness [14], hygroscopicity, and dimensional stability [16], etc.

The wear resistance of wood is its capacity to resist the action of external forces that cause mechanical damage to its surface. There are different types of wear, among which abrasion is the most interesting wear mechanism when the use of wood is considered [15]. Abrasive wear is the removal of material from a surface by a rubbing action of hard particles and protuberances. It is most often expressed as the volume loss, but also as the mass loss when samples of the same density are concerned. Abrasion resistance depends on the type of wood, its molecular structure and chemical composition, density, microstructure orientation, moisture content, and surface treatment.

The abrasion resistance of different wood species has been tested by means of various methods [16,17,18,19,20], but there have only been a few abrasion resistance studies on subfossil wood [21,22]. Subfossil wood is often used for carving of attractive jewellery and manufacturing of valuable furniture, but it has been observed that it often causes blunting of tools due to the impregnation caused by lying in river sediment. The fact that studies on subfossil oak wear resistance are so scarce led to the idea that such studies could render some useful guidelines to people who deal with the processing of this type of wood, and inspired us to make some further research into the abrasive properties of this valuable, attractive, and quite rare material.

## 2. Materials and Methods

The research was performed on three samples of the pedunculate oak. All the three samples originate from the area along the Sava River, on the border of the Republic of Croatia and Bosnia and Herzegovina, within a 3 km radius (Figure 1).

Sample 1 is an approximately 70-year-old trunk cut from a forest near Štitar village (45.107301; 18.632835), and it represents a recent, normal-wood oak which was never deposited in the river. Its growth ring width was about 1.56 mm.

Samples 2 and 3 represent subfossil wood with a similar growth ring width (1.16 mm and 1.44 mm) and were retrieved from the Sava River bed in the area near Domaljevac village (45.074475; 18.588781) (Figure 1). Their age was determined by the ^14^C radioisotope method, and the results are shown in the Table 1.

The test pieces used for density and abrasion volume loss measurements were cut from a part of the heartwood which showed no cracks or any other defects.

### Abrasion Volume Loss Tests

The abrasion volume loss was determined for each oak sample in three characteristic sections: cross (C), radial (R), and tangential (T) (Figure 2).

The test pieces used were four-sided prisms, about 30 mm long, with a square cross section measuring 5 mm × 5 mm (Figure 3). This square section was used as the friction surface in the abrasion testing. For each characteristic section (C, R, and T) of Samples 1–3, 20 test pieces were prepared. Before the abrasion process took place, all the test pieces were acclimatised for a few weeks at room temperature. In the course of the research, the laboratory temperature was 25 °C and the relative humidity level was 56%.

A Taber abraser with a rotating abrasive disc of 125 mm diameter was used to determine the abrasive volume loss of the test pieces. The rotational speed was 1 rpm and the peripheral speed was 0.251 m/s. A constant force of 4.91 N was applied to the test pieces to keep them in contact with the abrasive grains throughout the test. During the abrasion testing, sawdust was removed by a vacuum cleaner. A schematic representation of the test is shown in Figure 4.

The test pieces were abrased by sandpaper containing silicon carbide abrasive grains of different sizes. On all sandpapers (P80, P120, P240, and P800), five test pieces of every sample (1–3) were examined for all three characteristic sections (C, R, and T). In total, 180 test pieces were tested. Table 2 shows the sandpapers used in the test, their designations, and average grain sizes.

The weight loss (Δm) of every test piece was measured after 60 grinding cycles and converted into the volume loss (ΔV). In order to calculate the mass loss (Δm_12_) and density (ρ_12_) at 12% moisture content, the moisture content of the samples was determined according to the ISO 13061-1:2014 standard [25]. The density ρ_12_ was calculated according to the ISO 13061-2:2014 standard [26]. The value of the volume loss is inversely proportional to the abrasive wear resistance value.

## 3. Results and Discussion

Table 3 shows the results of the density measurements at 12% moisture content and the mass loss after 60 grinding cycles with respect to the four grades of sandpaper and to the three characteristic sections.

Figure 5, Figure 6 and Figure 7 show that the abrasion volume loss of all the samples and on all sections grows simultaneously with the growth of the grains, up to a certain point. After that point, in most of the tested samples, there occurs a decrease in the abrasion volume loss despite the increase in the average size of the abrasive grains. This phenomenon is called a critical abrasive grain size and is also observed in metallic and other non-metallic engineering materials [27,28,29,30]. The values of the critical abrasive grain size are between 125 µm and 200 µm, which coincides with the range of values of the trachea diameter of the test samples (Figure 2). Ohtani and Kamasaki [19] observed this effect in different types of angiosperm plants; a high positive correlation between the trachea size and that of the abrasive grains was noted. The correlation between the pore size and the critical abrasive grain size is of key importance for the abrasive wear rate. This statement is supported by the results of research carried out on compressed sugi wood [18], where the study showed that by reducing porosity in compressed wood, the grit size effect is gradually lost.

In Figure 5, Figure 6 and Figure 7 one can see that, among all the samples tested, the cross-section samples exhibit the lowest abrasion volume loss, while the tangential and radial sections showed similar abrasion wear rates. The reason for the different abrasion resistances of the three sections is based on the fact that wood is an anisotropic material, the mechanical properties of which depend on the orientation of its body (Figure 2). In the cross section, the fibres and the parenchyma are oriented perpendicularly to the wear surface, and in this direction the wood has the highest wear resistance. In the tangential and radial sections, pores, fibres, and parenchyma are oriented parallel to the wear surface of the samples and perpendicular to the direction of wear. In previous studies, similar results were observed in the Agathis species, where the wear rate in the radial section was much higher than that in the cross section [31]. The study on the subfossil elm wood also showed that the cross section had the greatest abrasion resistance [32].

The differences in the wear rate between the cross section (C), the radial section (R), and the tangential section (T) of all the samples become larger as the abrasive grain size increases to the value of the critical abrasive grain size (Figure 5, Figure 6 and Figure 7). When the finest grit sandpaper is applied, the absolute difference in the abrasion volume loss between the three sections is not large. With the increase in the abrasive grain size to the critical value, the difference between the abrasion volume loss in the cross section and the other two sections increases in all the samples.

Table 4 shows the ratios between the mean abrasion volume loss in C, R, and T sections of all the samples and for all the grits of sandpaper. The results show uniformity of ratios in all the samples within the same grit of sandpaper.

## 4. Conclusions

The abrasive grain size has a great influence on the abrasive wear rate of all the samples in all the sections. As the average grain size increases, the abrasive wear rate increases simultaneously up to the point where the critical abrasive grain size is reached.

The phenomenon of critical grain size occurs in all the samples. The critical grain size for all the samples and in all characteristic sections ranges from 125 µm to 175 µm; this range corresponds to the average diameter of the pedunculate oak trachea.

The highest abrasion resistance was observed on recent oak.

A particular orientation of wood structure effects the abrasive wear rate. The cross section of the tested oak samples exhibited the highest resistance to the abrasive wear, while the tangential and radial sections showed similar resistance.

With the increase in abrasive grain size, the absolute difference in the abrasive volume loss rate between the cross section and the other two sections increases to a critical value in all samples.

The relative relations between the abrasive volume loss values in three different sections (C/R, C/T, and R/T) of all the samples are quite uniform.

In the future, it would be interesting to determine the parameters of the surface roughness of the machined wood surface and the type of abrasive wear, and to carry out the characterization of the abraded particles. All this would contribute to a broader picture of the research subject. The research could also be repeated on a few more different wood species in order to make comparisons between species with different microstructures.

## Figures and Tables

**Figure 1 materials-16-00432-f001:**
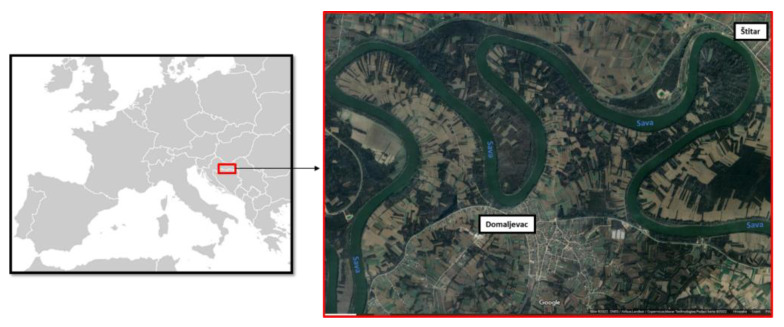
Locations of the origin of the samples (Wikimedia Commons 2020).

**Figure 2 materials-16-00432-f002:**
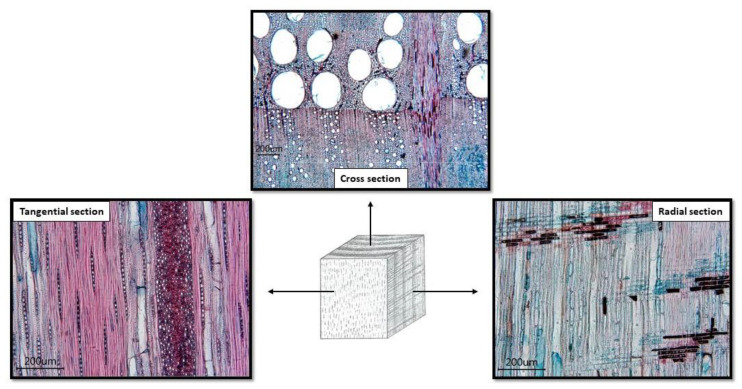
Oak microstructure in characteristic sections.

**Figure 3 materials-16-00432-f003:**
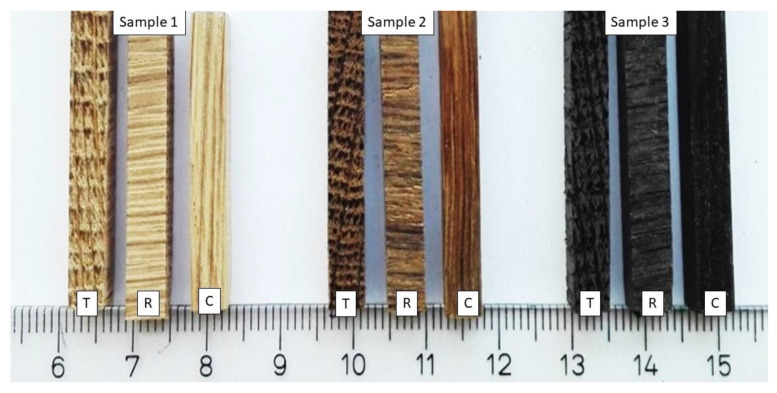
Test pieces prepared for testing.

**Figure 4 materials-16-00432-f004:**
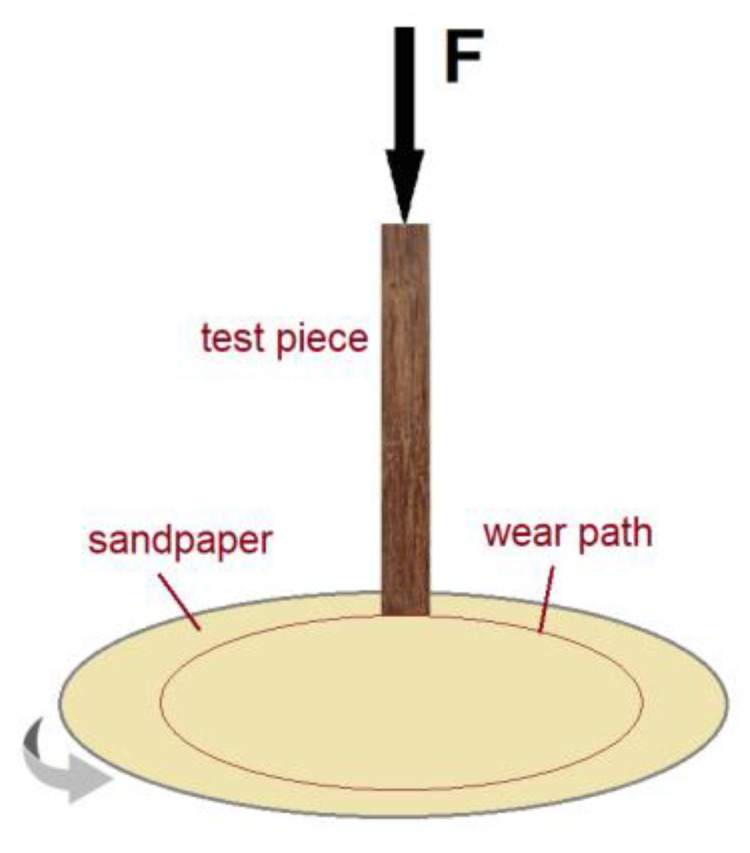
Schematic representation of the abrasive wear testing.

**Figure 5 materials-16-00432-f005:**
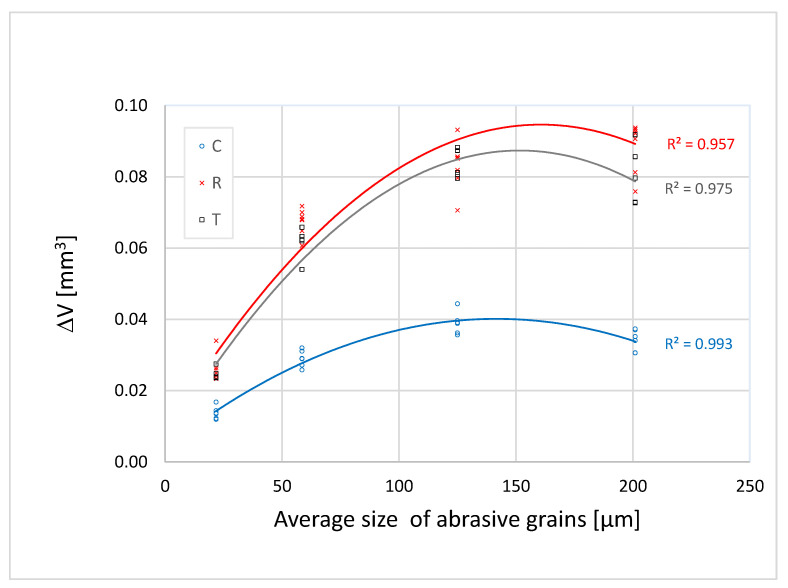
Dependence of the abrasion volume loss in Sample 1 on the average size of abrasive grains.

**Figure 6 materials-16-00432-f006:**
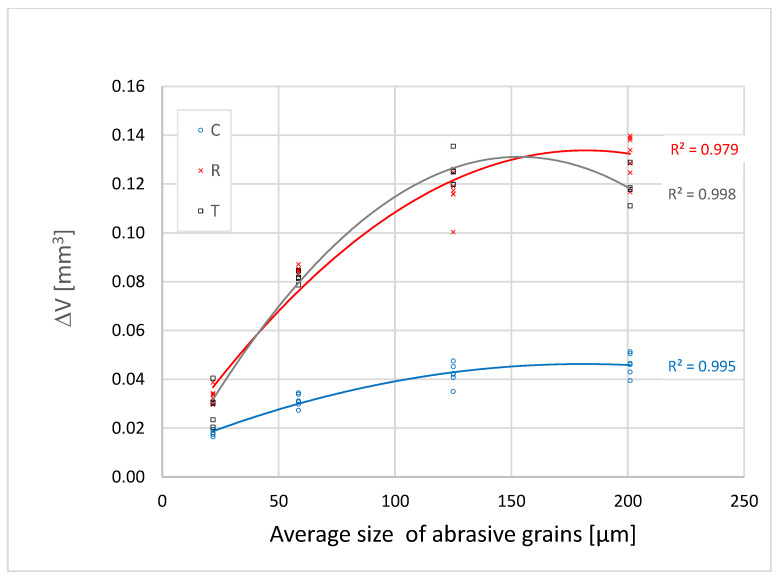
Dependence of the abrasion volume loss in Sample 2 on the average size of abrasive grains.

**Figure 7 materials-16-00432-f007:**
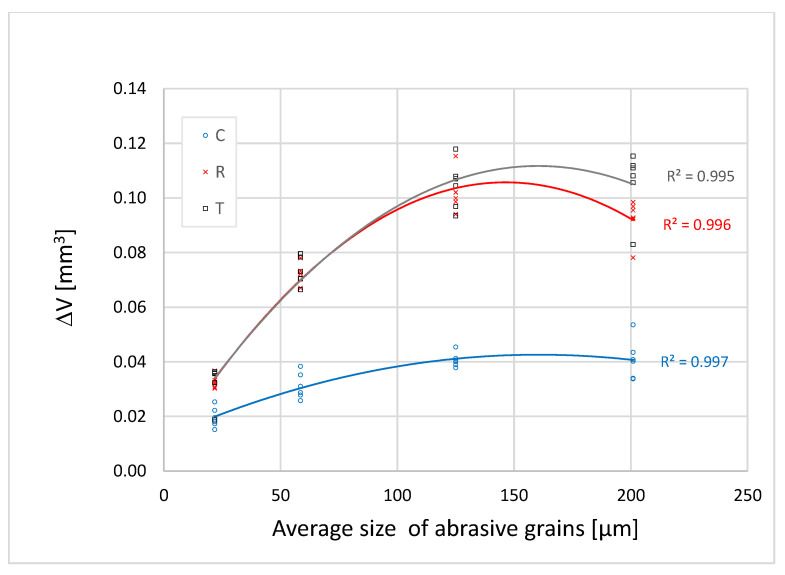
Dependence of the abrasion volume loss in Sample 3 on the average size of abrasive grains.

**Table 1 materials-16-00432-t001:** Age of tested subfossil samples.

	Conventional ^14^C Age (BP)/Years
Sample 2	1130 ± 50
Sample 3	1840 ± 55

**Table 2 materials-16-00432-t002:** Sandpapers used in testing.

Sandpaper Designation according to ISO [23,24]	Average Abrasive Grain Size [µm]
P 80	201
P 120	125
P 240	57
P 800	22.5

**Table 3 materials-16-00432-t003:** The density at 12% moisture content (ρ_12_) of oak wood samples and the mass loss (Δm) in three characteristic sections recorded on four grades of sandpaper (P80, P120, P240, P800).

		Δ_12_ (g/cm^3^)					At 12 % Moisture Content															
							Δm_800_ (g)				Δm_240_ (g)				Δm_120_ (g)				Δm_80_ (g)			
		Xp	SD	min	max	CV	Xp	SD	min	max	Xp	SD	min	max	Xp	SD	min	max	Xp	SD	min	max
	C	0.82	0.040	0.80	0.87	4.89	0.01	0.001	0.01	0.01	0.02	0.001	0.02	0.03	0.03	0.001	0.03	0.03	0.03	0.002	0.03	0.03
1	R	0.84	0.015	0.82	0.86	1.74	0.02	0.004	0.02	0.03	0.06	0.003	0.05	0.06	0.07	0.004	0.07	0.08	0.08	0.005	0.07	0.08
	T	0.86	0.016	0.84	0.88	1.91	0.02	0.001	0.02	0.02	0.05	0.004	0.05	0.06	0.07	0.006	0.06	0.08	0.07	0.006	0.06	0.08
	C	0.73	0.025	0.70	0.76	3.39	0.01	0.001	0.01	0.01	0.02	0.001	0.02	0.02	0.03	0.003	0.03	0.03	0.03	0.003	0.03	0.04
2	R	0.72	0.020	0.71	0.76	2.79	0.02	0.001	0.02	0.02	0.06	0.001	0.06	0.06	0.08	0.007	0.07	0.09	0.10	0.005	0.10	0.10
	T	0.71	0.023	0.69	0.73	3.25	0.02	0.006	0.01	0.03	0.06	0.002	0.06	0.06	0.09	0.004	0.08	0.09	0.08	0.006	0.08	0.09
	C	0.75	0.025	0.76	0.78	3.35	0.01	0.003	0.01	0.02	0.02	0.003	0.02	0.03	0.03	0.001	0.03	0.03	0.03	0.005	0.03	0.04
3	R	0.78	0.007	0.78	0.79	0.91	0.03	0.002	0.02	0.03	0.06	0.003	0.05	0.06	0.08	0.006	0.07	0.09	0.07	0.006	0.06	0.08
	T	0.74	0.018	0.72	0.77	2.47	0.02	0.005	0.01	0.03	0.05	0.004	0.05	0.06	0.08	0.006	0.07	0.09	0.08	0.008	0.06	0.08

Note: C—cross section, R—radial section, T—tangential section, Xp—mean value, SD—standard deviation, CV—coefficient of variation (%), min—minimum value, max—maximum value).

**Table 4 materials-16-00432-t004:** The values of the ratio between the volume loss of Samples 1–3 in the cross (C), radial (R), and tangential (T) sections for all grits of sandpapers.

Sample	Ratio	Average Size of Abrasive Grains (µm)			
		21.8	58.5	125.0	201.0
	C/R	0.5	0.4	0.5	0.4
**1**	C/T	0.6	0.5	0.5	0.4
	R/T	1.1	1.1	1.0	1.1
	C/R	0.6	0.4	0.4	0.3
**2**	C/T	0.6	0.4	0.3	0.4
	R/T	1.1	1.0	0.9	1.1
	C/R	0.6	0.4	0.4	0.4
**3**	C/T	0.6	0.4	0.4	0.4
	R/T	1.0	1.0	1.0	0.9

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
