# Peer review of "Effect of Abrasive Grain Size on the Abrasion Volume Loss of Subfossil and Recent Oak Wood in Three Characteristic Sections"

_materials, 2023, doi:10.3390/ma16010432_

Round 1

Reviewer 1 Report

The paper refers to the analysis of influence of grain size on the abrasion volume loss for two subfossil oak. Analyzing the results, it is highlighted that the average size of abrasive grains and the wood structure orientation have a influence on the abrasion volume loss. The material and method chapter are adequately described.

I recommend the following suggestions:

-          The abstract can be improved. It is difficult to clear understand the last paragraph (Line 21-26).

-          Line 18-19: it is mentioned that “…the average size of abrasive grains and 18 the orientation of the wood structure have a great influence on the abrasion volume loss.” In terms of statistically analysis, please specify the notion “grate influence”. Significant or not significant? It could be applied a statistically test to confirm the affirmation.

-          The resolution of Figure 2 is not very well. Could it be improved?

-          In Table 2 it is specified ISO/FEPA. This reference should also appear in the references section. The same situation for ISO 13061-1:2014 and ISO 13061-2:2014 standards (Line 103-104).

-          Line 120. It is omitted the unit “μm” for value 125 (“ … the critical abrasive grain size are between 125 and …”).

Author Response

Dear Reviewer 1,

Thank you for all your suggestions and comments. We have accepted most of them and we made suggested corrections to improve our paper. Please check the corrected manuscript (the Track changes tool was used) and the attached document with explanations.
Thank you in advance for your time!

Kind regrads,
Sara Essert

Reviewer 2 Report

This manuscript investigated the abrasive resistance of two different ages of subfossil oak wood and present oak wood. Since subfossil wood is rare and valuable, accumulation of basic properties of it is important. I have no objection to the experimental procedure and result itself, however the representation of the data could be improved. I would recommend this manuscript be accepted after minor revision.   P2 2. Materials and methods How many test pieces were prepared per condition?     P3 L85  56%-> 56% RH?   

Figure 3 

Samples prepared for testing -> Test pieces prepared for testing More information (Sample 1,2,3 and R,C,T) is needed in the figure.   Table 3 Please add the Sample 1, 2 and 3 on the left column.   Figure 5, 6 and 7 Plot of mean value with error bar for standard deviation might be better. If the main objective of this study is to compare the abrasion volume loss of 2 subfossil wood to recent wood, it might be better to show the figures with three different dimensions (i.e Figure 5:R , Figure 6:C and Figure 7:T)   P8L157 Table 4. shows -> Table 4 shows   P7L159-162 "The findings stated above point to the fact that the relative relations between the wear rate values in three different sections do not change significantly regardless of the absolute value of volume loss, size of abrasive grains, samples age, differences in their density or in the mean width of annual rings." If the author would like to say they do not change significantly, a multiple comparison among 3 samples within the same parameter and same grain size should be tested.

Author Response

Dear Reviewer 2,

Thank you for all your suggestions and comments. We have accepted most of them and we made suggested corrections to improve our paper. Please check the corrected manuscript (the Track changes tool was used) and the attached document with explanations.
Thank you in advance for your time!

Kind regrads,
Sara Essert

Reviewer 3 Report

The presented research the influence of the abrasive grain size of sandpaper on the abrasion volume loss of subfossil wood, a valuable and rare material often used for production of expensive and decorative artistic items. The results showed that the average size of abrasive grains and the orientation of the wood structure have a major influence on the abrasion volume loss. Additionally, the authors were observed phenomenon of the critical size of abrasive grains.

In my opinion, the paper is an interesting one, but my greatest doubt in the presented research is the lack of any evaluations of the quality of the machined surface, for example in the form of measurements of the roughness of the ground surface. The authors presenting only the results of abrasion mass/volume loss tests is only touching on the problem and I don't think it's appropriate to consider machining performance in isolation from the surface quality achieved.

I also found some errors in this manuscript, and it should be improved.

Weak

Some disadvantage of this paper is that it cites somewhat outdated sources. Only one cited paper is from the last 4 years.

Noticed errors

1.      Lines 41, 51. Bulk citation of more than 2-3 sources is not a good practice and citing more than ten literature sources together is unacceptable. Authors should expand the analysis of the state of the issue and refer to each cited item separately in a few sentences.

2.      At the end of Chapter 1 Introduction, the research gap should be more clearly highlighted.

3.      Throughout the paper, harmonize the symbol used to separate the integer part from the fractional part of a number written in decimal form. English uses a dot (full stop) rather than a comma.

4.      Figure 4 is of inadequate quality and needs improvement.

5.      In Figures 5-7, the data is presented quite clearly, but the identification of the lines themselves is poor, as the markers are poorly visible (small and hollow inside). To improve visibility, the markers should be slightly enlarged and filled with color. Changing the type of line to continuous would also help with distinction.

6.      Conclusions chapter. The conclusions in present form are poor and they should be extended, also, with the addition of conclusion(s) for further research.

7.      The cited literature list in several places’ deviates from the journal template. This should be corrected.

 Small errors

These errors do not diminish the value of this interesting work, but need to be improved

1.      Line 80. Is: 3 centimetres; should be: 3 centimeters (or better 30 mm)

2.      Line 93. The words below is redundant.

3.      Lines 101-102. Is: (Δm12) and density (ρ12); should be: (Δm12) and density (ρ12). It would be necessary to check the entire work and agree on the notation of expressions with subscripts.

4.  Line 143. Is: oriented perpendiculary, should be: oriented perpendicularly

5.      Line 150, in word crosss double s is enough.

Author Response

Dear Reviewer 3,

Thank you for all your suggestions and comments. We have accepted most of them and we made suggested corrections to improve our paper. Please check the corrected manuscript (the Track changes tool was used) and the attached document with explanations.
Thank you in advance for your time!

Kind regrads,
Sara Essert

Round 2

Reviewer 3 Report

The authors corrected most of my comments and the obstacles to publishing this paper no longer exist. In my opinion, the paper in its current form is suitable for publication.